

# Physical activity during pregnancy and its influence on delivery time: a randomized clinical trial

Raquel Rodríguez-Blanque[1,2], Juan Carlos Sánchez-García[2,3], Antonio Manuel Sánchez-López[2] and María José Aguilar-Cordero[1,2,3]

[1] Hospital Universitario San Cecilio, Granada, Spain
[2] Andalusia Research Plan, Research Group CTS 367, Granada, Spain
[3] Departamento de Enfermería/Facultad de Ciencias de la Salud, Universidad de Granada, Granada, Spain

Corresponding author
Juan Carlos Sánchez-García,
jsangar@ugr.es, jcsg750@gmail.com

## ABSTRACT

**Introduction**. During pregnancy, women often change their lifestyle for fear of harmful effects on the child or themselves. In this respect, many women reduce the amount of physical exercise they take, despite its beneficial effects.

**Objective**. To determine the duration of labor in pregnant women who completed a program of moderate physical exercise in water and subsequently presented eutocic birth.

**Methods**. A randomized trial was performed with 140 healthy pregnant women, divided into an exercise group (EG) ($n = 70$) and a control group (CG) ($n = 70$). The women who composed the study population were recruited at 12 weeks of gestation. The intervention program, termed SWEP (Study of Water Exercise during Pregnancy) began in week 20 of gestation and ended in week 37. Perinatal outcomes were determined by examining the corresponding partographs, recorded by the Maternity Service at the Granada University Hospital Complex.

**Results**. The intervention phase of the study took place from June through October 2016, with the 120 women finally included in EG and CG (60 in each group). At term, 63% of the women in EG and 56% of those in CG had a eutocic birth. The average total duration of labor was $389.33 \pm 216.18$ min for the women in EG and $561.30 \pm 199.94$ min for those in CG, a difference of approximately three hours ($p < 0.001$).

**Conclusions**. The women who exercised in water during their pregnancy presented a shorter duration of labor than those who did not. The difference was especially marked with respect to the duration of the first and second stages of labor.

## INTRODUCTION

Although regular physical exercise is known to produce many health benefits, at all stages of life, doubts have been expressed regarding its appropriateness during pregnancy, and about the type, frequency, intensity and duration of such exercise (*ACOG, 2015*).

During pregnancy, the body undergoes many changes, primarily affecting the loco-motor system, and exercise routines must be modified. However, lack of knowledge in

this respect often makes health care personnel extremely cautious in their recommendations (*Sui, Turnbull & Dodd, 2013*).

Sedentarism and obesity are aspects of a major public health problem, one that also affects women of childbearing age. During pregnancy, many women are strongly motivated to make changes in their lifestyle. This intention, together with the provision of appropriate information and advice from health care professionals, will help them acquire habits beneficial to their own health and that of their babies.

Recent studies have shown that exercise during pregnancy prevents excessive weight gain (*Pelaez, 2011*), gestational diabetes (*Barakat et al., 2013*), and high blood pressure (*Genest et al., 2012*). It benefits not only the mother but also the fetus, since it reduces the risk of macrosomia (*Barakat et al., 2013*) and lowers the risk of premature birth (*Da Silva et al., 2017*). All of these factors promote physiological delivery (*Barakat et al., 2012*).

Exercising in water during pregnancy offers many advantages. For example, submergence in water decreases body weight and facilitates movement, thus preventing overload on the joints and back. Furthermore, exercising in water enables the woman to focus on breathing—rhythm, phases, volume, and type of respiration. These benefits are relevant to the subsequent stages of labor and delivery (*Castillo-Obeso, 2002*).

We hypothesize that the practice of moderate physical exercise in water, following the guidelines of the SWEP (Study of Water Exercise during Pregnancy) method, will both improve aerobic capacity and also strengthen the muscles involved in childbirth (those of the abdomen and lower back), thus enhancing their performance during the second stage of labor.

### Study aim

To determine the duration of labor in pregnant women who completed a program of moderate physical exercise in water and subsequently presented eutocic birth.

## MATERIAL AND METHOD

An open-label randomized controlled trial was conducted, in which both the subjects and the investigators were aware of the intervention. This trial complied with the CONSORT standards published in 2010 (*CONSORT, 2010*). It was approved by the Research Ethics Committee for the province of Granada, and assigned File No. SWEP-13-06. At all times, the study was conducted in accordance with the provisions of the Declaration of Helsinki, as amended at the 64th WMA General Assembly, Fortaleza, Brazil, October 2013. Written informed consent was obtained from all participants.

The trial is registered at the US National Institutes of Health (ClinicalTrials.gov), under the title "Physical Activity in Pregnancy and Postpartum Period, Effects on Women", Number NCT02761967.

### Participants

The women who took part in this study were recruited at the Clinical Management Unit in La Zubia, which belongs to the Metropolitan Health Care District of Granada, Spain. This Unit has five health centers and an outpatients' clinic. In Spain, woman of

childbearing age with amenorrhea normally attend their local health center to diagnose pregnancy. At the centers in question, potential recruits were informed about the study and those who expressed interest were asked to provide contact details. Subsequently, the midwife responsible for the recruitment of study participants telephoned to arrange an appointment at the patient's ultrasound clinic, during week 12 of gestation (these appointments took place in March and April 2016).

Contact was made with 364 potential participants. Of these women, 224 were excluded from consideration for inclusion in the study: 122 did not meet the inclusion criteria, 82 refused to participate (no reasons stated) and 20 cited personal reasons, such as fear of physical exercise during pregnancy, family responsibilities or lack of time due to their employment.

The final sample, thus, consisted of 140 pregnant women, aged between 21 and 43 years, who were divided into two subgroups: Exercise Group (EG) and Control Group (CG), with 70 women in each.

The following inclusion conditions were applied for participation in the study: good health, with an uncomplicated singleton gestation; presenting none of the absolute contraindications described in Box 1 of ACOG document number 650 (*ACOG, 2015*), including heart disease, restrictive lung disease, cervical incompetence, risk of preterm birth, second or third-trimester persistent bleeding, preterm labor during current pregnancy, rupture of membranes, preeclampsia or pregnancy-induced hypertension or severe anemia. In the case of a relative contraindication, a favorable report from the patient's obstetrician was required for participation in the study. If warning signs were observed, suggesting that physical exercise should be suspended during pregnancy, the patient's gynecologist was consulted about the advisability of continuing with the program.

As the study data were obtained from the University Hospital Complex of Granada, Spain, giving birth in a different hospital was a further criterion for exclusion.

The data for the participants were obtained at the Health Centers of the Metropolitan Health District of Granada, Spain. The clinical data on the birth process—labor and birth—were obtained from the partograph, a graphic record of the evolution of labor obtained from the mother's clinical history, held at the hospital. The province of Granada has two public hospitals for maternal and child health care. Both apply the same health care policies for childbirth, in accordance with published guidelines for normal delivery (*Ministerio de Sanidad, Seguridad Social e Igualdad, 2017*). The protocols for health care during childbirth can be consulted via the following links:

San Cecilio University Hospital:

https://www.husc.es/especialidades/ginecologia_y_obstetricia/guias_y_protocolos_asistenciales.

Virgen de las Nieves University Hospital:

https://www.huvn.es/asistencia_sanitaria/ginecologia_y_obstetricia.

## Intervention

The women in EG performed moderate physical exercise in water, following the SWEP guidelines. This exercise provides two main benefits; it improves aerobic capacity and it strengthens the muscles especially involved in childbirth, as well as those in the abdomen and lower back. This results in better muscle tone in the areas of the body most subject to tension during pregnancy, and is excellent preparation for labor, generating increased lung capacity and greater strength and resilience of the muscles involved in each stage of childbirth (*Aguilar-Cordero et al., 2016*). The SWEP method is applied from weeks 20 to 37 of pregnancy and consists of three 60-minute sessions per week, each with 45 min' activity followed by 15 min' relaxation. The sessions are based on three phases: first, the warm-up, followed by the main phase, with aerobic movements and strength-endurance exercises, designed specifically for pregnant women, and a final phase of stretching and relaxation. The sessions take place in the mornings, after adequate caloric intake and hydration.

The warm-up phase is firstly general, performed out of the water, and then specific, in the water and with exercises appropriate to those which will follow. The aerobic element of the exercise is designed to be performed in a 25-meter pool where the women can perform swimming exercises adapted to the corresponding phase of their pregnancy. The strength and resistance exercises are then carried out in a pool 10 m long and 1.50 m deep, and are also adapted to the women's stage of pregnancy (*Aguilar-Cordero et al., 2016*).

The women in CG followed the usual recommendations during pregnancy, which included general guidance from the midwife about the positive effects of physical exercise. The participants in CG had regular meetings with health providers (midwives, obstetricians, and family physicians) during their pregnancy, as did those in EG.

## Expected results and study instruments
### Sociodemographic and anthropometric variables
Age, height, weight during first and third trimesters, weight gain and parity.

### Physical effort exerted and level of intensity
The perceived effort and intensity of exercise were measured using the Borg Rating of Perceived Exertion (*Borg, 1982*) (according to which a score of 12–14 is classed as "somewhat hard") in order to ensure the moderate nature of the exercise performed, which at all times was in accordance with ACOG recommendations (*ACOG, 2015*).

The participants' heart rate during the training sessions was monitored using a Quirumed OXYM2000 portable pulse oximeter, to measure pulse and oxygen saturation. Heart rate was determined at the end of each exercise session for the women who produced a value >14 on the Borg Scale.

### Perinatal results
A graphic record of the evolution of labor in each case was obtained by means of a partograph (*De Groof et al., 1995*; *Lennox, Kwast & Farley, 1998*; *Napoles et al., 2004*; *Tinker & Koblinsky, 1994*; *Walraven, 1994*; *WHO Maternal Health and Safe Motherhood, 1994*). The partograph was used to determine the following variables: administration of

oxytocin, anesthesia during labor (epidural or subdural), duration of gestation (days), neonatal birthweight, skin-skin contact and duration of the first, second, and third stages of labor. The first stage of labor is the time elapsed from 4 to 10 cm. of dilation; the second stage is the time elapsed from then until full dilation is reached and fetal expulsion occurs; and the third stage is the time elapsed from then until expulsion of the placenta and the membranes. In our analysis, the weeks of gestation were transformed into days of gestation until the delivery took place.

The type of birth was also noted, but only in order to determine the percentages of eutocic, instrumental or cesarean birth. In our calculation of the duration of labor, only births that ended spontaneously (i.e., excluding instrumental births and cesarean sections) were taken into account. The duration of labor was defined as the sum of the durations of the first, second, and third stages defined above.

## Sample size

The necessary sample size was calculated taking into account the previous findings of *Barakat, Lucia & Ruiz (2009)*, who also studied the impact of a program of physical exercise for women during the second and third trimesters of pregnancy and who highlighted the benefits of participation in such a program, for the mother and for the child. In order to obtain a statistical power of 80% to detect differences in the test of the null hypothesis $H_0: \mu_1 = \mu_2$, using Student's $t$-test for two independent samples, with a 5% level of significance and a joint standard deviation of 2.67, we calculated that a sample of at least 136 women would be needed, with 68 in each study group.

## Randomization

The sample allocation procedure was randomized as follows: each woman who arrived at the health center and met the inclusion criteria was assigned a ticket bearing a serial number, by the researcher responsible for recruitment. The ticket was then placed in an opaque envelope, and the envelope in a container. When all the envelopes were in the container, the Principal Investigator removed 70 envelopes, which were assigned to EG, and the remaining 70 were assigned to CG.

## Statistical analysis

The normality of the distribution of the numerical variables was tested by the Kolmogorov–Smirnov method. The remaining study variables were qualitative and were studied by a descriptive analysis.

When the study variable was continuous and normally distributed, Student's $t$-test was used to determine whether the inter-group differences were significant. For the qualitative variables, the chi-square test was applied to determine the differences between the groups. The statistical power of the study for the temporal variables was 82%. Multiple linear regression models were obtained for the first and second stages and for the total duration of delivery.

All statistical analyses were performed using IBM SPSS 19 statistical software (SPSS Inc., Chicago, IL, USA). The significance level was set at $p < 0.05$.

## RESULTS

Figure 1 shows the flow diagram of the sample selection procedure used.

As shown in Fig. 1, five women in EG and six in EG had to leave the study due to pathologies such as the risk of premature birth, or delayed intrauterine growth, pregnancy-induced hypertension, or premature rupture of the membranes.

Table 1 shows that EG and CG presented similar populational characteristics, with no significant differences in age, weight, or height at baseline, or in parity (Student's $t$-test; $p > 0.05$). Neither were there any statistically significant differences in the administration of oxytocin in order to induce labor (Student's $t$-test; $p = 0.390$). The use of anesthesia was similar in both groups (Student's $t$-test; $p = 0.092$).

There were no significant differences between EG and CG in the reason for hospital admission (Pearson's chi-square: $p = 0.776$), and in both groups a high percentage of women were admitted with the diagnosis of latent phase of labor (EG 58.5% vs. CG 64.1%).

The intensity of the physical exercise was monitored on a daily basis; the women were asked to indicate their perceived level of exercise intensity, according to the Borg scale. The aim was to maintain a daily score ranging between 12 and 14 ("somewhat hard" on the Borg scale) and corresponding to moderate intensity.

The onset of labor was more likely to be spontaneous in EG than in CG (70.8% versus 60.9%, respectively; $p > 0.05$), and was less likely to be induced (21.5% vs. 29.7%, respectively, $p > 0.05$). The main reason for the induction of labor was premature rupture of the membranes at term.

The duration of gestation in EG and CG was examined to determine whether physical exercise during pregnancy produced any alteration in this respect. No significant differences were observed between EG (281 days [277–286.50]) and CG (281 days [275.25–286.75]) ($p = 0.996$) (see Fig. 2). However, neonatal birth weight was significantly lower in EG than in CG ($p = 0.011$).

The total duration of labor was calculated as the sum of the first, second and third stages of labor, expressed in minutes. Significant differences were observed in this respect ($p < 0.001$) (see Table 2 and Fig. 3).

In stages 1 and 2, there was a difference of 2 h 25 min between EG (4 h 20 min) and CG (6 h 45 min) ($p < 0.001$). The difference of just over one hour for the second stage was statistically significant ($p = 0.007$). However, for the third stage there were no statistically significant differences, and both groups presented similar times. The total delivery time for EG was almost three hours less than that for CG.

Table 3 shows the results of the multivariate regression models for the first and second stages of labor, and for the total time of delivery.

The variables that influenced delivery times were induced labor, the use of epidural/subdural analgesia, the administration of oxytocin, and the performance of physical exercise. The duration of dilation was on average 88.87 min greater in CG than in EG, adjusting for all other variables, and 35.83 min greater until expulsion. The total delivery time was on average 139.13 min less in EG, adjusting for all other variables. The $R^2$ value
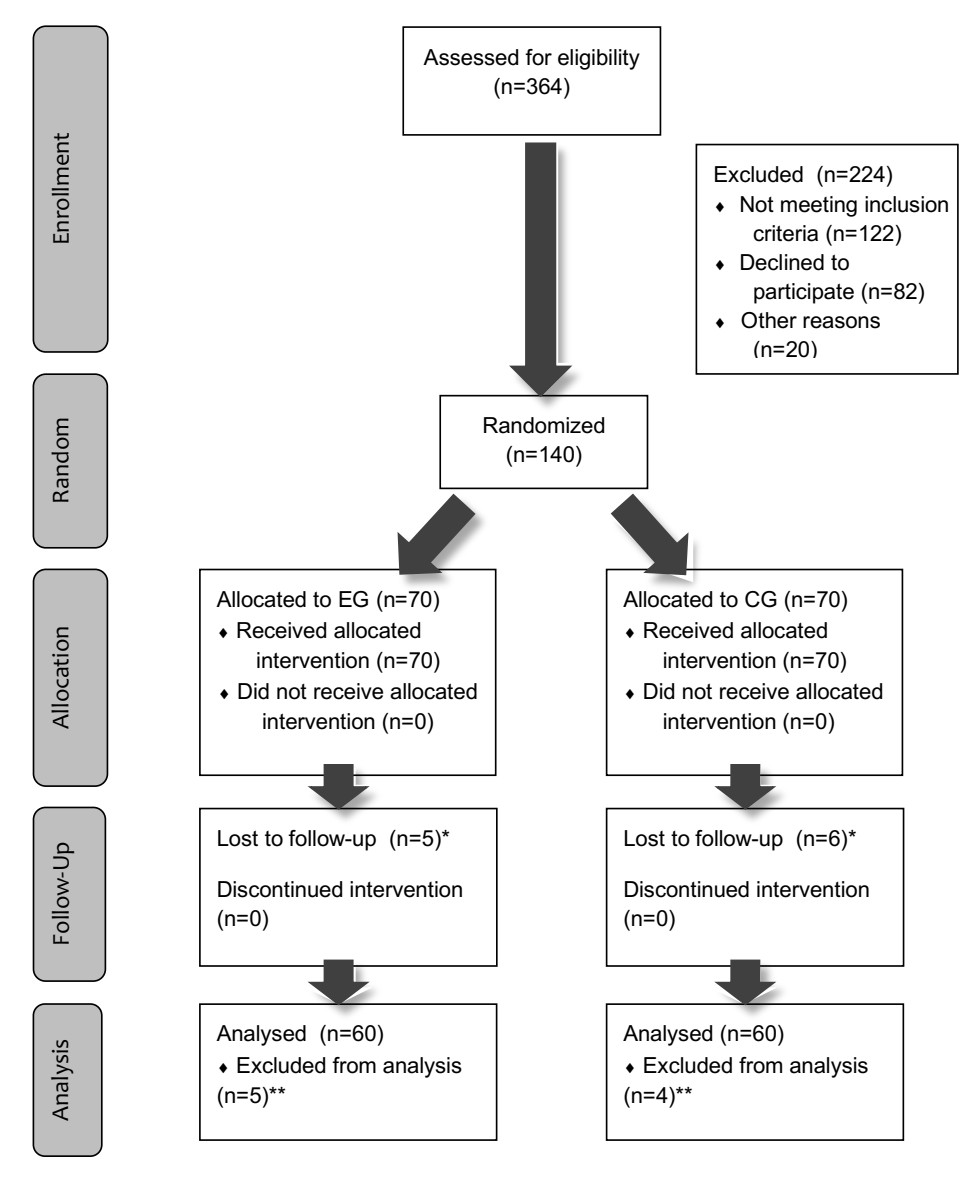

* **Complications during pregnancy:**
  **EG; 1 DIG, 1 TPB, 1P-IH, 2 PRM**
  **CG: 2 P-IH, 1 DIG, 1 TPB, 2 PRM**
** **Instrumental delivery and cesarean section**

DIG: Delayed Intrauterine Growth; TPB: Threat of Premature Birth; P-IH: Pregnancy-Induced Hypertension; PRM: Premature Rupture of Membranes.

**Figure 1  Flow diagram.**

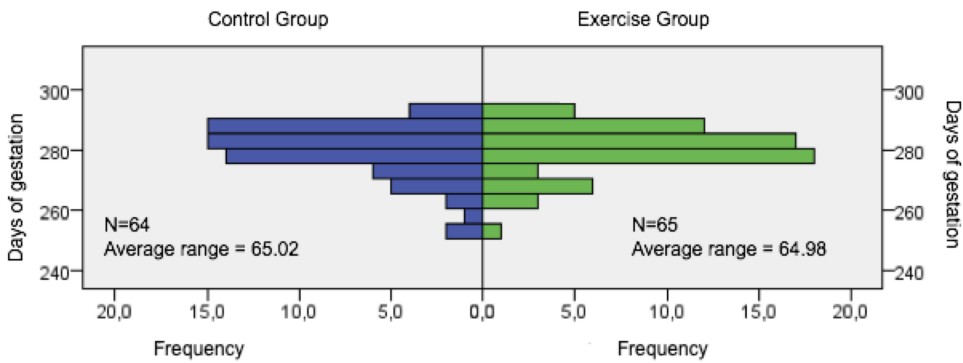

**Figure 2  Days of gestation.**

**Table 1  Baseline characteristics of the sample.**

| Characteristic | Exercise group $n = 65$ | Control group $n = 64$ | p-value |
|---|---|---|---|
| Maternal age, years (Mean ± SD) | 32.12 ± 4.43 | 30.58 ± 4.75 | 0.331 |
| Min–max | 21–43 | 22–43 | |
| Height (Mean ± SD) | 1.646 ± 0.06 | 1.651 ± 0.05 | 0.604 |
| Weight first trimester (Mean ± SD) | 67.07 ± 12.23 | 67.89 ± 12.58 | 0.71 |
| Weight third trimester (Mean ± SD) | 75.35 ± 12.13 | 79.05 ± 11.64 | 0.079 |
| BMI first trimester (Median [Q1–Q3]) | 23.89[21.52–27.51] | 24.01[21.78–26.58] | 0.953 |
| BMI third trimester (Mean ± SD) | 27.76 ± 4.03 | 29.03 ± 4.45 | 0.092 |
| Multiparous n (%) | 20 (30.77%) | 17 (26.56%) | 0.739 |
| Oxytocin n (%) | 19 (29.7%) | 14 (21.5%) | 0.39 |
| Anesthesia (Epidural/Subdural) n (%) | 55 (85.9%) | 47 (72.3%) | 0.092 |
| Duration of gestation (Mean ± SD) | 280.09 ± 8.26 | 279.70 ± 8.92 | 0.996 |
| Neonatal birthweight (Mean ± SD) | 3,259.00 ± 564.40 | 3,477.11 ± 414.51 | 0.011 |
| Skin-Skin contact n (%) | 53 (81.5%) | 51 (79.7%) | 0.790 |

**Table 2  Duration of labor (minutes).**

| | CG $n = 60$ | EG $n = 60$ | p-value |
|---|---|---|---|
| | **Median [Q1–Q3][a]** | **Median [Q1–Q3][a]** | |
| 1st stage | 405.00 [295.00–498.75] | 260.00 [137.50–390.00] | **<0.001** |
| 2nd stage | 152.50 [70.00–210.00] | 90.00 [30.00–187.50] | **0.007** |
| 3rd stage | 8.00 [5.00–10.00] | 5.00 [5.00–10.00] | 0.383 |
| | **Mean ± SD[b]** | **Mean ± SD[b]** | |
| Duration of labor | 561.30 ± 199.94 | 389.33 ± 216.18 | **<0.001** |

**Notes.**
[a] Median [Q1–Q3]: Median [Quartile 1–Quartile 3].
[b] Mean ± SD: Mean ± Standard Deviation.

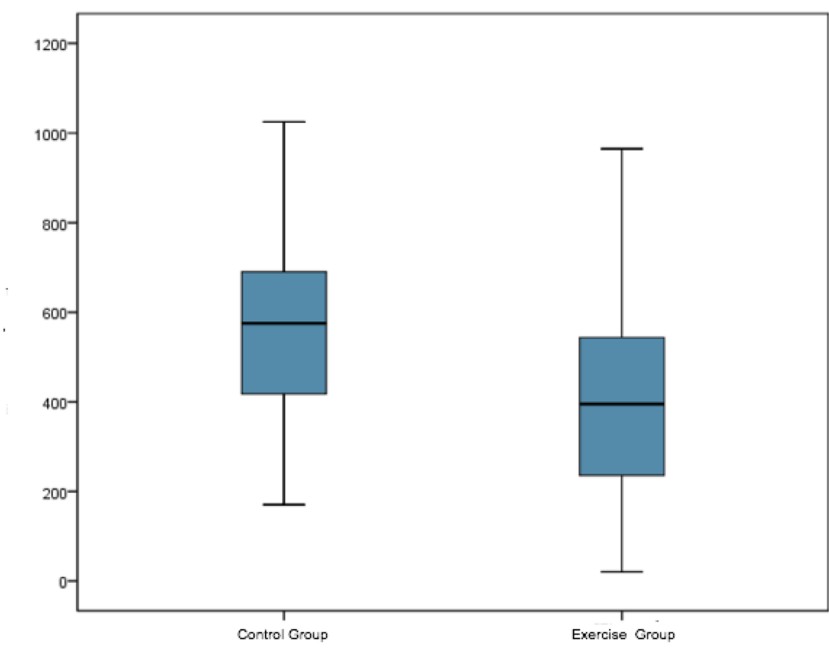

**Figure 3  Total duration (minutes) of labour-delivery in the study groups.**

for this model was 0.40, which indicates that these four variables accounted for 40% of the total variability of delivery duration.

## DISCUSSION

The first and second stages of labor were shorter for women who performed moderate physical exercise in water, following the SWEP guidelines, than for those who did not. The strengths of this study are the large number of participants, the high rate of follow-up, and the fact that the exercise program used (the SWEP method) was specially designed for pregnant women. Among its limitations are the difficulty of recruiting suitable participants, arising in part from the lack of information available in the health services to resolve doubts concerning physical exercise during pregnancy. Another limitation is that, in the linear regression model, in which some of the study hypotheses were not met, no residual normality was detected for the times of dilation, expulsion, and total delivery. Furthermore, the study population included only women who were not at risk during pregnancy, and therefore our results cannot be extrapolated to groups that include women with risky pregnancies.

*Salvesen et al. (2014)* conducted a study with similar characteristics, with a study population of 855 pregnant women who performed aerobic and strength-building exercises from weeks 20 to 36 of their pregnancy. This program consisted of a weekly group session led by a physiotherapist. The women were encouraged to conduct a 45-minute program at home, at least twice weekly, and to record this activity in a personal training diary. The authors concluded that the performance of physical exercise did

**Table 3** Regression models; first stage, second stage and total time of delivery.

| | $\beta$ | Standard error | P | 95% CI |
|---|---|---|---|---|
| **Regression model for time of the first stage (min)** | | | | |
| (Constant) | 9.767 | 68.016 | .886 | (−125.05,144.59) |
| Induced start | 105.111 | 32.217 | .001 | (41.25, 168.97) |
| Epidural analgesia | 149.135 | 36.752 | .000 | (76.29, 221.98) |
| Exercise group | 88.870 | 29.088 | .003 | (31.21, 146.52) |
| **Regression model for time of the second stage (min)** | | | | |
| (Constant) | .609 | 28.179 | .983 | (−55.22, 56.43) |
| Epidural analgesia | 64.787 | 15.629 | .000 | (33.82, 95.75) |
| Exercise Group | 35.833 | 13.211 | .008 | (9.66, 62.01) |
| **Regression model for total time of delivery (min)** | | | | |
| (Constant) | −27.412 | 70.228 | .697 | (−166.55, 111.72) |
| Induced start | 119.548 | 37.605 | .002 | (45.05, 194.05) |
| Epidural analgesia | 221.990 | 39.406 | .000 | (143.92, 300.06) |
| Exercise group | 139.133 | 32.685 | .000 | (74.38, 203.89) |
| Oxitocyn | 96.937 | 54.799 | .080 | (−11.63, 205.50) |

not influence the duration of the stages of birth, possibly because the exercise was not supervised at all times by a professional, and so the correct execution of the procedure could not be evaluated.

In our study, neonatal birthweight was significantly lower in EG than in CG ($p = 0.011$), which is consistent with the findings of *Barakat et al. (2010)* that physical exercise during pregnancy tends to reduce the birthweight, but has no influence on gestational age at birth.

However, *Perales et al. (2016)*, in a study with 166 pregnant women (83 in EG and 83 in CG), with an average age of 31.6 (SD 3.80) years, and who presented an uncomplicated singleton pregnancy, reported that participation in a physical exercise program during pregnancy is associated with a shorter first stage, with no significant differences in the duration of the second and third stages. This contrasts somewhat with the results of our study, according to which women who perform moderate physical exercise in water during the second and third trimesters of pregnancy present a significantly shorter first and second stage duration, with no significant differences in that of the third stage.

*Da Silveira & De Segre (2012)* carried out a prospective study of 66 pregnant women, aged between 18 and 30 years, with 37 in EG and 29 in CG. The women in EG performed moderate exercise twice a week for 50 min from week 20 of gestation until birth. These authors concluded that taking part in an exercise program during pregnancy influenced the type of birth, increasing the rate of vaginal deliveries. Similar findings were obtained by *Poyatos-León et al. (2015)*, who conducted a meta-analysis in 2015 and reported that regular exercise during pregnancy seemed to increase the likelihood of healthy pregnant women achieving a eutocic birth. However, this conclusion was not corroborated by our results, according to which the rates of spontaneous deliveries in CG and EG were similar (56.25% vs. 63.07%), as was that of cesarean section (14.06% vs. 12.30%).

*Barakat et al. (2009)* conducted a study of 160 pregnant women, 80 in each group. Those in EG performed 26 weeks of moderate-intensity exercises, with three sessions per week, beginning at week 12–13 of gestation and ending at week 38–39 of gestation. These authors reported that resistance training at moderate intensity, performed during the second and third trimesters of pregnancy, does not affect the type of birth, a finding corroborated by our own results. However, these authors did not obtain satisfactory results for the duration of the first and second stages of birth. This study, in which statistically significant differences were obtained between EG and CG, had a similar design to our own, the main difference between the two being that in ours the exercises were performed in water, and the training program used, the SWEP method, was designed especially for this study.

Our study shows that moderate, supervised resistance exercise does not endanger the health status of healthy pregnant women or that of the fetus, which is in accordance with the conclusions of *Petrov-Fieril, Glantz & Fagevik Olsen (2015)*, in whose study the women in EG performed a supervised resistance exercise program, twice weekly for 12 weeks (from week 14 to week 25 of gestation), with a moderate to vigorous activity level.

## CONCLUSIONS

Moderate physical exercise in water is associated with a reduced total time of labor and birth. In our study, the first and second stages of labor were significantly shorter in EG. Moreover, this activity increases the rate of eutocic birth, which enables the mother to recover more quickly and to make rapid skin-to-skin contact with the baby. In vaginal and instrumental deliveries, early skin-to-skin contact is sometimes delayed, when examination by the neonatologist is required, which is why spontaneous deliveries are more likely to be associated with rapid skin-to-skin contact. This, in turn, facilitates immediate breastfeeding.

As possible areas for future research, it would be interesting to investigate the use of this type of therapy in relation to health-related quality of life in healthy pregnant women and to consider how, during the postpartum period, it might influence quality of life, postpartum depression, postpartum fatigue, stress urinary incontinence, and abdominal diastasis. It could also be useful to study the economic impact of applying this type of therapy during pregnancy and the puerperium, in terms of reducing the need for medical consultations during pregnancy, with their associated costs, in comparison with the cost of implementing the program through healthcare services.

### Funding

No public funds were received for this study. The University of Granada collaborated by facilitating the use of aquatic resources at the School of Sports Science. The funders had no role in study design, data collection and analysis, decision to publish, or preparation of the manuscript.

## Grant Disclosures

The following grant information was disclosed by the authors:
School of Sports Science.

## Competing Interests

The authors declare there are no competing interests.

## Author Contributions

- Raquel Rodríguez-Blanque and Juan Carlos Sánchez-García conceived and designed the experiments, performed the experiments, analyzed the data, contributed reagents/materials/analysis tools, prepared figures and/or tables, authored or reviewed drafts of the paper, approved the final draft.
- Antonio Manuel Sánchez-López performed the experiments, contributed reagents/-materials/analysis tools, authored or reviewed drafts of the paper, approved the final draft.
- María José Aguilar-Cordero conceived and designed the experiments, authored or reviewed drafts of the paper, approved the final draft.

## Clinical Trial Ethics

The following information was supplied relating to ethical approvals (i.e., approving body and any reference numbers):

It was approved by the Research Ethics Committee for the province of Granada (File No. SWEP-13-06).

## Data Availability

The raw measurements are available in the Supplemental File.

## Clinical Trial Registration

The following information was supplied regarding Clinical Trial registration:

NCT02761967.

## Supplemental Information

Supplemental information for this article can be found online at http://dx.doi.org/10.7717/peerj.6370#supplemental-information.

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
