# Peer review of "Physical activity during pregnancy and its influence on delivery time: a randomized clinical trial"

_PeerJ, doi:10.7717/peerj.6370_

## Round 0.1 · original submission · Major Revisions

Please, address all the reviewers' observations and it would be advisable to check the language by a fluent-English speaker.

Reviewer 1 ·

Basic reporting

Please see the uploaded review document

Experimental design

Please see the uploaded review document

Validity of the findings

Please see the uploaded review document

Additional comments

Please see the uploaded review document

Annotated reviews are not available for download in order to protect the identity of reviewers who chose to remain anonymous.

·

Basic reporting

The objective of the paper is to evaluate the effect of water exercise on labour duration.
The topic is relevant and the role of “movement” in pregnancy and birth not negligible.

Experimental design

The study is well described in the method section and the characteristics of exercise well standardized, however the section regarding results shows major limits and it is lacking regarding “obstetric” points.
Power analysis is not reported.
In particular because the variability of the duration of labour is enormous, influenced by an innumerable number of factors, and the diagnosis of the beginning of labour is not easy, the definition of the beginning of labor and the cm of dilation at admission to hospital and at the beginning of partograph compilation should be reported.
Otherwise the conclusion could be completely (and anyway interesting) different: a woman performing water exercise spends less time in labour in Hospital respect to a woman not performing such exercise

Validity of the findings

First question: all the deliveries were at the same hospital? In hospital?
I cannot understand the total number of CS or operative vaginal delivery, how many elective and how many in labor? How many induced? How many admitted for labor? Reason for induction?
The overall percentage of CS of the Hospital/s? Underline eventual difference that could be due to different setting
Which is the meaning of : “birth in progress” , and At the beginning of labor, the rate of spontaneous birth was higher in EG than in CG (70.8% versus 60.9%, respectively)
Data on neonatal outcome should be reported, neonatal weight is another important potential confounder present in the dataset, include in the results.

The sentences reported below are redundant and the definition of duration in days obvious: report only the duration of gestation in days
The duration of gestation in EG and CG was examined to determine whether physical
exercise during pregnancy produced any alteration in this respect. To perform this comparison, the duration of gestation (in weeks) recorded in the partograph was converted to days of gestation. The resulting distribution did not differ significantly (p=0.996) between EG (281 days [277 - 286.50]) and CG (281 days [275.25 - 286.75]) (see Figure 2).

Conclusions regarding neonatal outcome should be supported by results, otherwise removed

Statistical analysis should be supervised
English should be reviewed

Reviewer 3 ·

Basic reporting

The data are presented in a confused way. In particular, there is no correspondence between the numbers shown in the abstract and those reported in the results.

Experimental design

The experimental design is correct and appropriate but the evaluation criteria of the results have to be defined more.

Validity of the findings

The conclusions is welcome, but they should be certain.

Additional comments

The study of physical exercise during pregnancy is an important topic which has not been explored enough. Therefore, this paper could make a contribution. However, the results could be imprecise because the Authors did not define the first stage of labour accurately. The first stage of labour is divided into the latent phase and the active phase. The length of the latent phase varies physiologically from pregnant woman to pregnant woman and is difficult to define. I suggest the Authors review the results taking into considerations the active phase of the first stage of labour and identifying an objective point in the active phase, such as 4 cm or 6 cm cervical dilatation. More precise data are required to demonstrate that physical exercise significantly reduces the length of labour, in order to be sure that these results are not due to physiological variations.

---

## Round 0.2 · accepted · Accept

All the reviewers' observations have been adequately addressed. Please note that reviewer #2 has noted that there is a sentence reported in figure 1 that needs to be translated. Congratulations!

Reviewer 1 ·

Basic reporting

The referee feels that the authors have appropriately addressed the questions raised during the reviewing process in the new version of their manuscript and in the rebuttal letter.

Experimental design

see above

Validity of the findings

see above

Additional comments

see above

Reviewer 3 ·

Basic reporting

no comment

Experimental design

no comment

Validity of the findings

no comment

Additional comments

The study of physical exercise during pregnancy is an important topic, which has not been explored enough.
In this new rewrite, the authors have more accurately specified the different stages of labor. Now the results are consistent and deserving to be published.
In Figure 1 to translate the sentence “Complicaciones durante la gestación” into English